# Predictive Potential of RNA Polymerase B (II) Subunit 1 (RPB1) Cytoplasmic Aggregation for Neoadjuvant Chemotherapy Failure

**DOI:** 10.3390/ijms242115869

**Published:** 2023-11-01

**Authors:** Bence Nagy-Mikó, Orsolya Németh-Szatmári, Réka Faragó-Mészáros, Aliz Csókási, Bence Bognár, Nóra Ördög, Barbara N. Borsos, Hajnalka Majoros, Zsuzsanna Ujfaludi, Orsolya Oláh-Németh, Aliz Nikolényi, Ágnes Dobi, Renáta Kószó, Dóra Sántha, György Lázár, Zsolt Simonka, Attila Paszt, Katalin Ormándi, Tibor Pankotai, Imre M. Boros, Zoltán Villányi, András Vörös

**Affiliations:** 1Department of Biochemistry and Molecular Biology, University of Szeged, 52 Középfasor, H-6726 Szeged, Hungary; 2Department of Pathology, Albert Szent-Györgyi Medical School, University of Szeged, Állomás utca 1, H-6725 Szeged, Hungary; 3Competence Centre of the Life Sciences Cluster of the Centre of Excellence for Interdisciplinary Research, Development and Innovation, University of Szeged, Dugonics tér 13, H-6720 Szeged, Hungary; 4Department of Oncotherapy, Albert Szent-Györgyi Health Centre, University of Szeged, 12 Korányi Fasor, H-6720 Szeged, Hungary; 5Department of Surgery, Albert Szent-Györgyi Health Centre, University of Szeged, 8 Semmelweis Street, H-6725 Szeged, Hungary; 6Department of Radiology, Albert Szent-Györgyi Health Centre, University of Szeged, 6 Semmelweis Street, H-6725 Szeged, Hungary; 7Genome Integrity and DNA Repair Core Group, Hungarian Centre of Excellence for Molecular Medicine (HCEMM), University of Szeged, Budapesti út 9, H-6728 Szeged, Hungary

**Keywords:** RPB1, RNAPII, CCR4-NOT, assemblysomes, condensates, aggregation, epirubicin, neoadjuvant therapy, invasive carcinoma of no special type

## Abstract

We aimed to investigate the contribution of co-translational protein aggregation to the chemotherapy resistance of tumor cells. Increased co-translational protein aggregation reflects altered translation regulation that may have the potential to buffer transcription under genotoxic stress. As an indicator for such an event, we followed the cytoplasmic aggregation of RPB1, the aggregation-prone largest subunit of RNA polymerase II, in biopsy samples taken from patients with invasive carcinoma of no special type. RPB1 frequently aggregates co-translationally in the absence of proper HSP90 chaperone function or in ribosome mutant cells as revealed formerly in yeast. We found that cytoplasmic foci of RPB1 occur in larger sizes in tumors that showed no regression after therapy. Based on these results, we propose that monitoring the cytoplasmic aggregation of RPB1 may be suitable for determining—from biopsy samples taken before treatment—the effectiveness of neoadjuvant chemotherapy.

## 1. Introduction

Neoadjuvant chemotherapy is a common approach for treating breast cancer [1]. The predictive markers used today (Estrogen Receptor, Progesterone Receptor, Human Epidermal Growth Factor Receptor 2, and the Marker of Proliferation Ki-67) are very useful for selecting the appropriate drugs in each case; however, there is a lack of predictive markers for the expected outcome of the selected therapy. Epirubicin, an anthracycline topoisomerase II inhibitor that acts as a DNA-intercalating agent, thereby inhibiting both the transcription and replication processes, is a widely administered drug in neoadjuvant chemotherapy [2]. Based on the effectiveness of the drug, tumors can be classified into the following three main groups: complete, partial, and non-regressive, respectively. One possible explanation for the resistance to epirubicin is that cells rely on existing mRNA depositories under genotoxic stress when transcription is blocked. Stress granules and P-bodies are phase-separated membraneless organelles that, together with other RNA–protein complexes, play important roles in the stress response of cells and maintain a transcription-independent source of mRNAs [3]. The recently discovered NOT1-containing assemblysomes are another type of these RNP granules with a potential role in the response of tumor cells to therapy [4,5].

Assemblysomes may play a role in stress response by supporting the co-translational assembly of stress-responsive protein complexes. One already confirmed complex dependent on assemblysomes for its integrity is the 26S proteasome, but according to recent in silico predictions, complexes involved in DNA damage response are also regulated by assemblysomes [4,5]. Assemblysomes contain the NOT1 subunit of the CCR4–NOT complex, the major deadenylase of eukaryotic cells [4]. NOT1 has been recently reported to function as a chaperone platform (for reviews see [6,7,8]). The impairment of the chaperone capacity of the cell leads to failure in the assembly of protein complexes. Consequently, the loss of function of complexes playing a role in apoptosis or cellular immune response processes can be beneficial for tumor progression. RPB1 is the largest subunit of the eukaryotic RNA polymerase II (RNAPII) and—as revealed in yeast—is a highly aggregation-prone protein that depends both on the CCR4–NOT complex and the chaperone HSP90 for its native conformation [9]. RNAPII is a 12-subunit complex. It assembles in the cytoplasm from RPB1 and RPB2 sub-complexes [10]. We hypothesized in a former study that if the CCR4–NOT complex plays a role in RPB1 and RPB2 sub-complex assembly, than generating a condition where both a CCR4–NOT subunit and RPB2 are limiting, will lead to RPB1 accumulation in the cytoplasm as only the fully assembled RNAPII can bind to Iwr1 that facilitates the nuclear import of the assembled complex [11]. According to our former results, ovaries of NOT3 and RPB2 trans-heterozygous *Drosophila melanogaster* accumulated RPB1 foci in the cytoplasm, revealing not only the role of the CCR4–NOT complex in RNAPII assembly but also that RPB1 aggregation during CCR4–NOT-mediated co-translational RNAPII assembly is a conserved phenomenon [9].

Invasive breast carcinoma of no special type (NST)—previously known as invasive ductal carcinoma—is the most common type of breast cancer. It is an infiltrating and malignant proliferation of neoplastic cells in the breast tissues. Tumors belonging to the non-regressive group in their response to neoadjuvant chemotherapy are resistant toward the toxic transcription inhibitory effect of epirubicin. We considered the possibility that the reason behind the phenomenon is that epirubicin-resistant cells are able to compensate for the loss of transcription by buffering their gene expression relying on mRNA deposited in condensates such as P-bodies, stress granules, or NOT1-containing assemblysomes. Since it has been recently reported that these cytoplasmic bodies co-localize with, or in the case of assemblysomes, their formation is even dependent on NOT1 [4,12], it is reasonable to assume that increased P-body, stress granule, and assemblysome biogenesis may generate a condition where soluble NOT1 becomes limiting. This may lead at least partially to the loss of function of NOT1 in processes, including its chaperone platform role in the correct folding of RPB1 [9]. Considering that RPB1 is a component of RNAPII, the major machinery of transcription, it may be even permitted to lose significant active RPB1 in the form of aggregates in tumor cell clones that otherwise tolerate the complete blockade of transcription by epirubicin. Therefore, RPB1 cytoplasmic aggregation might be a potent indicator of tumor cell clones in biopsies that will tolerate transcription blockers used as chemotherapeutic agents.

In the present work, we investigated the role of co-translational protein aggregation, particularly the cytoplasmic aggregation of RPB1, in tumor cells from biopsy samples of patients with invasive carcinoma of NST to determine its potential as a predictor of chemotherapy resistance.

## 2. Results

### 2.1. Large RPB1 Cytoplasmic Foci Are Apparent in Non-Regressive Invasive Carcinoma of NST Cells

We aimed to investigate the contribution of co-translational protein aggregation to the chemoresistance of invasive carcinoma of NST. Therefore, immunofluorescent microscopy was performed to follow the cytoplasmic aggregation of RPB1, the largest subunit of RNAPII in biopsy samples, taken from patients before the administration of neoadjuvant chemotherapy (Figure 1 and Appendix A). Compared with patients who responded to the therapy with either partial or complete tumor regression, those with tumors showing no regression in tumor size following chemotherapy had numerous cytoplasmic RPB1-aggregated foci in their first biopsy samples.

As a control to test our sample preparation and staining protocol, we followed a modified nuclear RPB1. The fifth serine at the C-terminal heptapeptide-repeated domain of RPB1 becomes phosphorylated at promoters by the basal transcription factor TFIIH [13]. We could detect only nuclear RPB1, as expected, with the phospho-5-serine-modified RPB1-recognizing antibody in all three phenotypic categories (Appendix A), clarifying that the cytoplasmic staining is not an artifact of sample preparation, fixation, or staining.

After investigating the samples with known regression phenotypes, we tested the use of the RPB1 cytoplasmic phenotype screen in predicting the outcome of therapy by categorizing samples into the three possible outcome groups in a blind experiment. Of the 13 investigated samples, we have managed to correctly categorize 10 (Appendix A). The prediction accuracy of no regression, partial regression, and total regression cases was above 84%, 76%, and 92%, respectively, as calculated according to [14] (Appendix A). The Matthews correlation coefficient (MCC) describes the relationship between the classification models as a perfect classification (which has a MCC of 1), random guessing (which has a MCC of 0), and a total disagreement, meaning there are only false positives and negatives (which has a MCC of −1) [15]. The MCC of the prediction of no regression, partial regression, and total regression cases was 0.675, 0.5, and 0.8216, respectively, giving credit to the classification method based on following RPB1 cytoplasmic aggregation phenotypes (Appendix A).

### 2.2. Cytoplasmic RPB1 Foci Are Apparent Sporadically in Renal-Cell Carcinoma Cells, Although Not in the Cells of Surrounding Tissues

We investigated another type of cancer for a similar phenotype to determine if the presence of cytoplasmic RPB1 foci is a specific phenotype observable only in invasive carcinoma of NST. We chose to study clear-cell renal-cell carcinoma samples from patients who underwent radical or partial nephrectomy; the samples had been previously analyzed using fluorescent microscopy (Figure 2, Appendix A) [16].

Here, we investigated both the malignant cells and the surgically removed surrounding healthy tissue. According to our results, RPB1 cytoplasmic foci were observable in some clear-cell renal carcinoma cells, although not in surrounding healthy renal tissue. Thus, we clarified that RPB1 cytoplasmic accumulation is not specifically observed in invasive carcinoma of NST; however, based on the analysis of 30 patients and 100 cells, it appears to be tumor-specific in renal tissues. Cytoplasmic RPB1 foci were discovered in the tumor samples of 18 patients in the cohort.

## 3. Discussion

Reoccurring failure in protein synthesis can lead to the loss of function of specific tumor-suppressor proteins. It is easy to see that a mutation affecting a ribosomal protein (RP) that is situated exactly where the correct interaction with the tRNA, for example, the decoding of the mRNA, normally takes place can lead to such a failure and to the formation of aberrant or non-functional proteins. RPL5 and RPL10 are located in sites where decoding takes place inside the ribosome, and exhibit mutations with considerable frequency in tumor cells [17]. A recent study has reported that circulating tumor cells isolated from patients with breast cancer exhibited increased ribosome biogenesis (RiBi) and RP translation, pointing toward an altered function of their translation machinery [18]. NOT1 of the CCR4–NOT complex is associated with RiBi and RP mRNAs, facilitating their translation in yeast [19,20]. Furthermore, NOT1 has been shown to become engaged with stress granules, and is important in the formation of assemblysome and mRNA condensates [4,12]. Cytoplasmic mRNA-containing condensates represent the reservoirs of transcription-independently expressible gene products. One possible explanation of how tumor cells become resistant to transcription blockade by epirubicin is that they contain stored mRNA in condensates acting as a reservoir to maintain gene expression; simultaneously, mRNA decay initiated by the major deadenylase CCR4 function is altered in them. Consequently, the loss of transcription is buffered by other gene expression processes. Even the loss of protein degradation may be beneficial if the transcription is limiting. Remarkably, the CCR4–NOT complex contains the major deadenylase of eukaryotic cells and, as a chaperone platform, it has been revealed to be involved in the assembly of RNAPII and proteasome [4,9]. Although it remains elusive, it is tempting to assume that the frequent aggregation of RPB1 in tumor cells resistant to the transcription inhibitor epirubicin may be a consequence of a failure in the CCR4–NOT-dependent co-translational folding of RPB1. Notably, other functions of the CCR4–NOT complex that may become more pronounced in tumor cells at the expense of its role in RNAPII assembly are exactly roles that have the potential to supplement the loss of transcription, including its role in mRNA condensate formation and RP and RiBi translation [12,19]. Increased RiBi provides the basic machinery for translation, which has the potential to buffer the loss of the other major gene expression process under epirubicin-induced transcription stress.

Considering its potential predictive significance, we find it urgent to report that, based on our results, and the blind test predicting no regression cases above an 84% accuracy, RPB1 has a tendency to aggregate in epirubicin-resistant tumor cells. The phenotype is easy to follow from biopsy samples immediately after cancer diagnosis. Unfortunately, a reliable quantitative analysis was not possible in terms of the cytoplasmic aggregation of the RBP1 protein. However, it is essential to highlight that, in the current clinical practice, the diagnosing pathologists must base their assessment on what they can observe in biopsy samples.

We propose that patients with cytoplasmic RPB1 foci in their biopsy samples should consider the high risk of ineffective, but toxic treatment with transcription blockers along with the time loss, and choose surgery instead of neoadjuvant chemotherapy. Alternatively, we suggest the use of proteasome inhibitors instead of transcription blockers in these specific cases, as an enhanced protein aggregation elevates the burden on the ubiquitin–proteasome system. We propose that blocking the proteasome might lead to amino acid scarcity in tumor cells where RPB1 aggregation indicates defective translation.

Moreover, we report that RPB1 aggregation is not invasive-carcinoma-NST-specific; however, it may be ubiquitous as we detected this phenotype in 18 patients in a cohort of 30 diagnosed with clear-cell renal-cell carcinoma. Indeed, A-431 epidermoid carcinoma, U-251 glioblastoma, and also U2OS osteosarcoma cell lines accumulate cytoplasmic RPB1 condensates in their cytoplasm according to the Human Protein Atlas [21]. Interestingly, cytoplasmic staining was not detected in an NIH/3T3 normal mouse fibroblast cell line of non-tumorous origin [21]. This fact, and also that the phenotype was not apparent in the healthy surrounding tissue in any of the surgically removed renal samples, indicate that RPB1 cytoplasmic aggregation might be a consequence of an oncogenic mutation rather than present before the malignant transformation in the healthy renal tissue of susceptible patients.

Our study’s ability to categorize cases without relying on extensive quantitative analysis, which might not be possible in a clinical setting, underlines the potential practical utility of this screening methodology in identifying non-regressive cases. Nevertheless, the collection of more data and further investigation are needed to strengthen our findings.

## 4. Materials and Methods

### 4.1. Cohort Selection

A total of 24 patients diagnosed with invasive carcinoma of no special type (NST), and 30 patients diagnosed with clear-cell renal-cell carcinoma, who underwent radical or partial nephrectomy at the University of Szeged, were selected for this study. The clinical and anthropometric characteristics of the patients whose samples are analyzed in the present study are listed in Appendix A.

### 4.2. Preparations of Normal and Tumorous Tissues

For immunofluorescence staining, the tissues were embedded in Shandon Cryomatrix gel (Thermo Fisher Scientific, Budapest, Hungary), and 5 μm sections were prepared on Superfrost Ultra Plus slides by cryostat (Cryostar NX50, Thermo Fischer Scientific). The samples originated from the tumor and from intact kidney parenchyma which was far away from the original tumor (1–2 cm distance). Grossing and blanking were performed by an experienced genitourinary pathologist.

### 4.3. Preparations of Formalin-Fixed Paraffin-Embedded (FFPE) Biopsy Samples

Tissues were fixed with 10% formalin for 24–48 h at room temperature. The process of paraffin embedding used ethanol + IPA (70–96%) and xylene/IPA, following standard protocols. The biopsy samples were embedded in paraffin blocks, and 5 μm sections were prepared using microtome following the protocol described in [16,22].

### 4.4. Immunostaining of Frozen Tissues

Tissues were fixed for 10 min using acetone and subsequently washed three times with PBS solution. Subsequently, the tissues were permeabilized with 0.3% Triton-X-100/PBS (Molar Chemicals Ltd., Halásztelek, Hungary) for 20 min at 25 °C. The sections were subsequently blocked with 5% BSA (Thermo Fisher Scientific)/0.3% Triton X-100/PBS (Molar Chemicals Ltd.) for 1 h. The samples were incubated with primary antibodies diluted in 1% BSA/PBST: anti-RNAPII (1BP7G5 from L. Tora, IGBMC, France) in 1:250 dilution. After the washing steps, secondary GAM Alexa 488 (Thermo Fisher Scientific, A11029) and antibody in 1:500 dilution were used. Finally, the cells were mounted with DAPI-containing ProLong Gold Antifade Reagent (Thermo Fisher Scientific). The samples were visualized using FLUOVIEW FV10i (Olympus, Budapest, Hungary) confocal microscopy. The same exposition time was used for the capturing of every image.

### 4.5. Immunostaining of FFPE Samples

Paraffin was removed by soaking the slides for 10 min in xylene. Rehydrating was performed by soaking the slides in sequentially decreasing *v/v*% ethanol (100%, 95%, 70%, and 50%) for 2 min each. The samples were permeabilized for 15 min in 0.1% TritonX-100 (Molar Chemicals Ltd.) at room temperature. Three washing steps were performed with PBS, 30 s each. Blocking was performed in 5% BSA/PBS for 1 h. Primary anti-RNAPII (Abcam, Cambridge, UK, ab26721 or Santa-Cruz, Santa Cruz, CA, USA, sc-47701) and primary anti-HSP90 (Santa-Cruz, sc-13119) were used in 1:200 dilution in 5% BSA/PBS. After the washing steps, secondary GAM Alexa 647 (Abcam, ab150115) or GAR Alexa 488 (Abcam, ab150077) in 1:1000 dilution was used. The samples were visualized with FLUOVIEW FV10i confocal microscopy. A total of 4–8 fields of 46,509 µm^2^ were analyzed from each biopsy sample.

### 4.6. Statistical Analysis

The accurate and erroneous predictions of the blind experiment were used to calculate the accuracy and error rate of classification of the cases into no regression, partial regression, and total regression categories according to [15]. To provide further evidence that the results of the classification process were highly significant we calculated the Matthews correlation coefficient (MCC) according to [16]. However, it is important to acknowledge that the study’s statistical power was constrained by the relatively small sample size of 13 patients diagnosed with carcinoma of NST in the blind experiment.

## Figures and Tables

**Figure 1 ijms-24-15869-f001:**
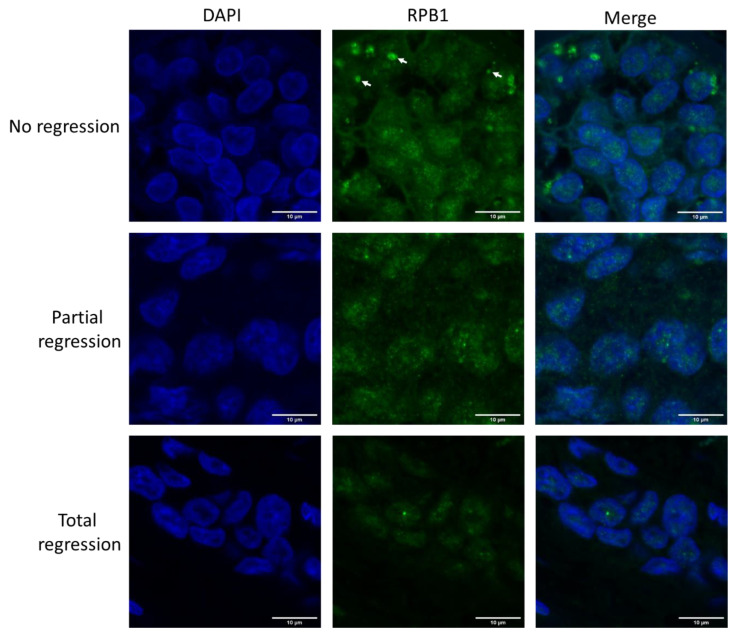
RPB1 appears in cytoplasmic foci in invasive carcinoma of no special type (NST) cells resistant to neoadjuvant chemotherapy. Immunofluorescent images of one representative case are shown from each of the three phenotypic categories as indicated on the left. In the cytoplasm of chemo resistant tumor cells (showing no regression after therapy), RPB1 foci are apparent (three examples are shown with white arrows). Nuclei are revealed using DAPI in cyan. RPB1 is shown in green, and the merged images of the two staining processes are highlighted. Scale bar: 10 μm.

**Figure 2 ijms-24-15869-f002:**
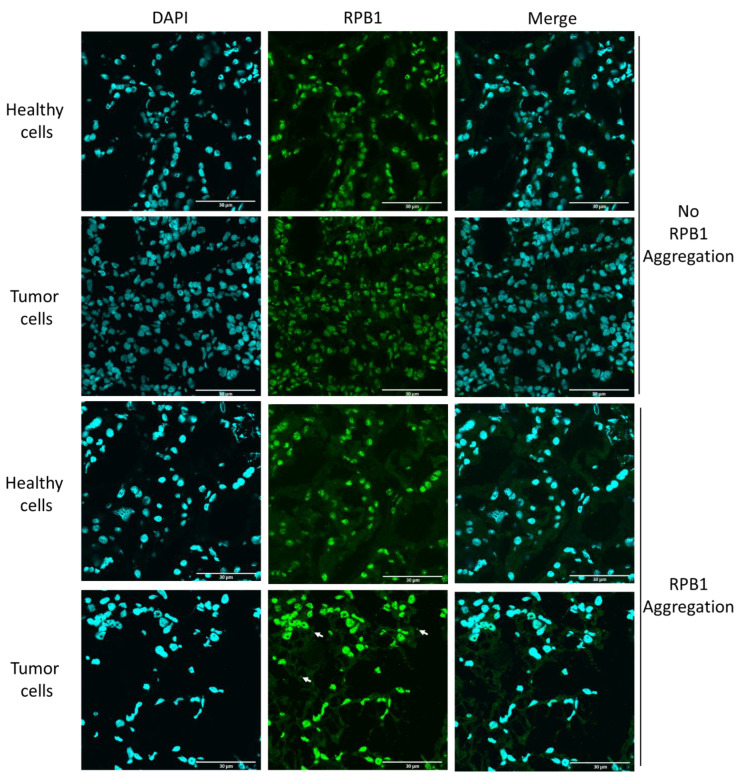
RPB1 aggregates sporadically in clear-cell renal-cell carcinoma. Fluorescent images of one representative case of clear-cell renal-cell carcinoma are shown, wherein RPB1 shows no cytoplasmic aggregation, and RPB1 foci are apparent (three examples are shown with white arrows). We have identified 18 patients where tumor-specific aggregation of RPB1 is detectable in surgically removed renal samples in a cohort of 30 patients (Appendix A). Nuclei are revealed using DAPI shown in cyan. RPB1 is shown in green, and the merged images of the two staining processes are highlighted. Scale bar: 30 μm.

## Data Availability

Data is contained within the article or Appendix A.

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
