# Peer review of "Predictive Potential of RNA Polymerase B (II) Subunit 1 (RPB1) Cytoplasmic Aggregation for Neoadjuvant Chemotherapy Failure"

_ijms, 2023, doi:10.3390/ijms242115869_

Round 1

Reviewer 1 Report

Comments and Suggestions for Authors

Present article by Miko et al titled "Cytoplasmic Aggregation of RPB1 Predicts Failure of Neoadjuvant Chemotherapy" needs following clarifications:

Please clarify the term used here "Carcinoma of No Special type". 

It would be nice if the last paragraph of introduction contains the statement of hypothesis. Like "The present manuscript what did the authors study etc.

Please provide the demographic details of the enrolled patients. 

Please do add a statement about informed consent.

The criteria for normal and tumor selection are not mentioned. What tissue was selected for normal. Was it adjacent normal or distant normal?

In Paragraph 2.3 please add a reference for the protocol used for paraffin embedding.

Paragraph 2.4 needs correction in formatting.

Please provide demographic details of enrolled patients.

The supplementary tables are less informative. Please provide additional details. 

Did you calculate statistical significance among the case/controls? In subgroup analysis the number of cases is very low and even zero. Therefore, drawing any conclusion based of this data will be difficult. However, the authors can add this as a limitation of this study. 

Reviewer 2 Report

Comments and Suggestions for Authors

In the present work Nagy-Miko et al. reported on the role of RPB1 cytoplasmic aggregation in the failure of neoadjuvant chemotherapy. Although the idea is interesting, the present work is poorly presented and the performed experiments do not support the conclusions. The authors although report on the collection of patient samples, the clinical and anthropometric characteristics are missing.

Second, their whole idea is based solely on qualitative criteria and therefore by no means their hypothesis can be supported. In order to support their hypothesis, the authors should have atleast perform protein analysis on cytoplasmic extracts, as well as evaluate the gene expressional profile of the molecules under investigation (e.g. RPB1 in the present case). 

Thus the present work is mere presentation of microscopy images with no other analysis or evaluation, and therefore any conclusions are vague. 

Overall, the present work does not have merit for publication.

Comments on the Quality of English Language

moderate English proof-editing.

Round 2

Reviewer 1 Report

Comments and Suggestions for Authors

The manuscript has been revised. I have no further comments.

Author Response

We sincerely thank once again the work of reviewer 1 in evaluating our manuscript.

In the name of all authors, with best regards,

Zoltan Villanyi

Reviewer 2 Report

Comments and Suggestions for Authors

I have read the author's response and although they mention that tissue samples are rare, this type of tumor is not. However, this is not the main issue here. The approach followed is not appropriate to claim the use of the methodology for diagnostic/prognostic purposes. Even if microscopy is the main focus, still there is significant information missing. How many fields were examined? quantification of fluorescence is missing. From the few microscopic images, how is to derive the conclusion that the presence of RPB1 plays a role in chemotherapy resistance. Obviously, it is not possible to address such a topic in its entirety however, there are many steps that can be undertaken as for example the investigation of other studies from publicly available data (ATLAS?) if this observation can be confirmed.

Further on, in order to claim that probably RPB1 and its cytoplasmic aggregation can play a role in chemotherapy resistance (or not) some quantification would be in place, with some statistical test to evaluate the hypothesis (regardless of the small sample size).
